# β-hydroxybutyrate accumulates in the rat heart during low-flow ischaemia with implications for functional recovery

Ross T Lindsay[1,2†*], Sophie Dieckmann[1], Dominika Krzyzanska[1], Dominic Manetta-Jones[1], James A West[2], Cecilia Castro[2], Julian L Griffin[2,3], Andrew J Murray[1]

[1]Department of Physiology, Development and Neuroscience, University of Cambridge, London, United Kingdom; [2]Department of Biochemistry and Cambridge Systems Biology Centre, University of Cambridge, London, United Kingdom; [3]Section of Biomolecular Medicine, Systems Medicine, Department of Metabolism, Digestion and Reproduction, Imperial College London, London, United Kingdom

*For correspondence:
rosstlindsay@gmail.com

Present address: †Research and Early Development, Cardiovascular, Renal and Metabolic Diseases, BioPharmaceuticals R&D, AstraZeneca Ltd, Cambridge, United Kingdom

**Abstract** Extrahepatic tissues which oxidise ketone bodies also have the capacity to accumulate them under particular conditions. We hypothesised that acetyl-coenzyme A (acetyl-CoA) accumulation and altered redox status during low-flow ischaemia would support ketone body production in the heart. Combining a Langendorff heart model of low-flow ischaemia/reperfusion with liquid chromatography coupled tandem mass spectrometry (LC-MS/MS), we show that β-hydroxybutyrate (β-OHB) accumulated in the ischaemic heart to 23.9 nmol/gww and was secreted into the coronary effluent. Sodium oxamate, a lactate dehydrogenase (LDH) inhibitor, increased ischaemic β-OHB levels 5.3-fold and slowed contractile recovery. Inhibition of β-hydroxy-β-methylglutaryl (HMG)-CoA synthase (HMGCS2) with hymeglusin lowered ischaemic β-OHB accumulation by 40%, despite increased flux through succinyl-CoA-3-oxaloacid CoA transferase (SCOT), resulting in greater contractile recovery. Hymeglusin also protected cardiac mitochondrial respiratory capacity during ischaemia/reperfusion. In conclusion, net ketone generation occurs in the heart under conditions of low-flow ischaemia. The process is driven by flux through both HMGCS2 and SCOT, and impacts on cardiac functional recovery from ischaemia/reperfusion.

## Introduction

Hepatic ketogenesis plays a vital role in starvation physiology, whereby acetyl-coenzyme A (acetyl-CoA) derived from fatty acid oxidation (FAO) undergoes a stepwise conversion to acetoacetate (AcAc) and β-hydroxybutyrate (β-OHB), which in turn act as fuels for the brain and other extrahepatic tissues (*Puchalska and Crawford, 2017*). Initially, mitochondrial thiolase activity catalyses the formation of acetoacetyl-CoA (AcAc-CoA) from two acetyl-CoA monomers, before a mitochondrial isoform of β-hydroxy-β-methylglutaryl (HMG)-CoA synthase (HMGCS2) catalyses the condensation of AcAc-CoA and a further acetyl-CoA to form HMG-CoA. This latter reaction is considered the rate-limiting step of ketogenesis, with the significant ketogenic capacity of liver attributed to the uniquely high activity of HMG-CoA synthase in comparison with other tissues (*McGarry and Foster, 1976*). In the mitochondria, HMG-CoA lyase converts HMG-CoA to AcAc, which in turn is reversibly converted to β-OHB under reducing conditions by β-hydroxybutyrate dehydrogenase (BDH1). In extrahepatic tissues (*Figure 1*), BDH1 catalyses the reverse reaction converting β-OHB to AcAc, from which AcAc-CoA is obtained by the exchange of a CoA through the action of succinyl-CoA:3-oxaloacid CoA transferase (SCOT). Mitochondrial thiolase activity then yields two acetyl-CoA molecules, which enter the Krebs cycle.

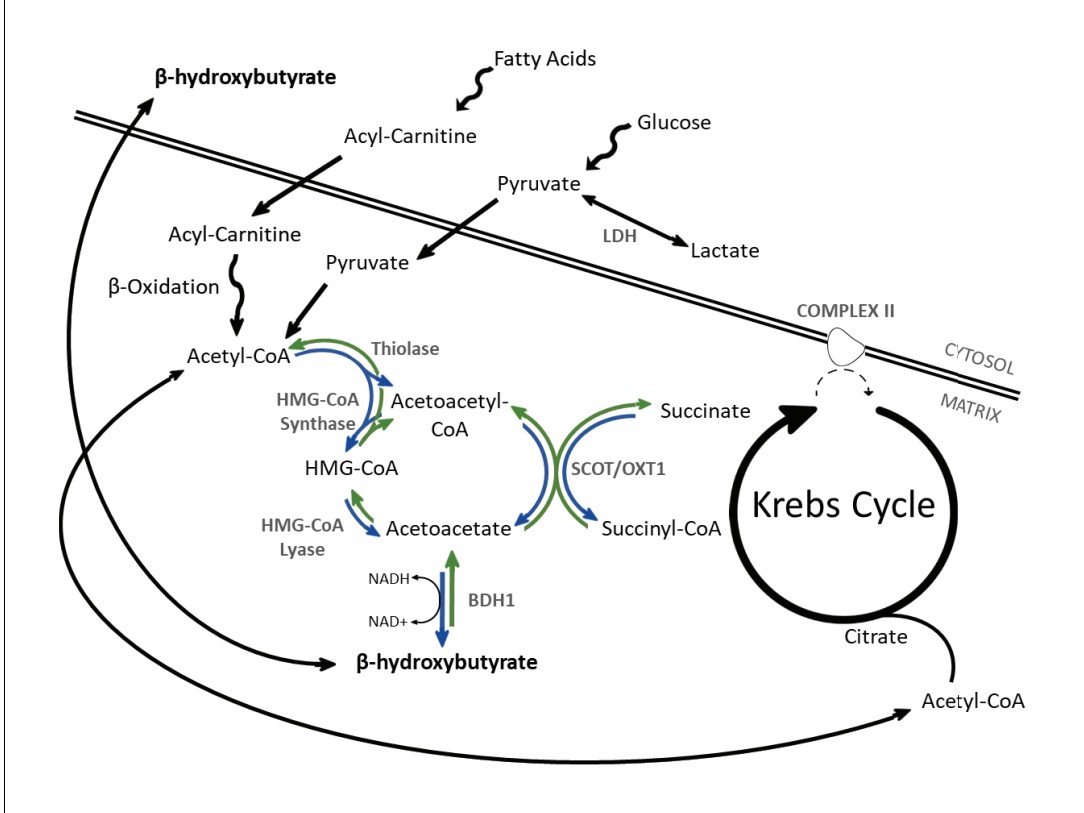

**Figure 1.** β-OHB metabolism. Catabolism of both fatty acids and glucose yields acetyl-coenzyme A (acetyl-CoA), following uptake to the mitochondria via acyl-carnitines and pyruvate, respectively. In the liver, ketogenesis involves the conversion of acetyl-CoA to acetoacetyl-CoA, β-hydroxy-β-methylglutaryl-CoA (HMG-CoA), acetoacetate, and then β-hydroxybutyrate in a series of reversible reactions. The enzymes which catalyse this are HMG-CoA synthase, HMG-CoA lyase, and β-hydroxybutyrate dehydrogenase (BDH1). In extrahepatic tissues, HMG-CoA can be bypassed through conversion of acetoacetate to acetoacetyl-CoA in conjunction with the conversion of succinyl-CoA to succinate, catalysed by succinyl-CoA-3-oxaloacid CoA transferase (SCOT). Rather than being oxidised in the mitochondria, under anaerobic conditions such as ischaemia, pyruvate can instead be converted to lactate by the enzyme lactate dehydrogenase (LDH). For bidirectional enzymes involved in both ketolysis and ketogenesis, the direction of ketogenic flux is represented by blue arrows, while ketolytic reactions are represented in green.

The heart has a significant capacity for ketone body oxidation (*Cotter et al., 2013*), but any extra-hepatic tissue which oxidises ketone bodies also has the capacity to generate ketone bodies (*Puchalska and Crawford, 2017*). This was shown to occur in the kidney through the reversible enzymatic activity of mitochondrial thiolase and SCOT, and the authors speculated that this may generally be the case for extrahepatic tissues (*Weidemann and Krebs, 1969*). In the isolated perfused rat heart, net production of AcAc and β-OHB has been observed when fatty acids are supplied via the perfusion medium (*Comte et al., 1997*; *Opie and Owen, 1975*). Inclusion of [1-$^{13}$C]octanoate in the medium and gas chromatography-mass spectrometry (GC-MS) analysis of ketone body enrichments in the coronary effluent implicated FAO as a source of this ketone body efflux (*Comte et al., 1997*), which may represent a spillover pathway when acetyl-CoA availability exceeds the capacity of the Krebs cycle. Independent of fatty acid provision, net production of AcAc from acetyl-CoA was seen in pig heart extracts but only in the presence of succinate (*Stern et al., 1953*), whilst AcAc and β-OHB synthesis was seen in isolated rat heart mitochondria respiring with pyruvate and malate (*LaNoue et al., 1970*).

The conversion of AcAc to β-OHB by BDH1 is dependent on NADH availability, and accordingly was found to be increased in the isolated perfused rat heart under anoxic conditions (*Opie and Owen, 1975*). Reduction of AcAc was seen in sheep heart homogenates treated with succinate and the mitochondrial complex I inhibitor amytal (*Krebs et al., 1961*; *Kulka et al., 1961*), and complete reduction of AcAc to β-OHB was reported in isolated rat heart mitochondria respiring with succinate (*Schönfeld et al., 2010*). We therefore hypothesised that net cardiac production of β-OHB can occur

under ischaemic conditions, when availability of NADH and acetyl-CoA would be increased and when succinate accumulates (*Chouchani et al., 2014*; *Laplante et al., 1997*; *Taegtmeyer, 1978*; *Zhang et al., 2018*).

## Results

### β-hydroxybutyrate accumulates in the ischaemic rat heart

The concentration of β-OHB was measured in the left ventricle of rat hearts snap-frozen pre-ischaemia, at the conclusion of the ischaemic period, and post-reperfusion, and in the coronary effluent throughout the perfusion protocol using liquid chromatography coupled mass spectrometry (LC-MS) (*Figure 2A–B*). The concentration of β-OHB in left ventricle after 32 min of aerobic perfusion was 0.2 nmol per mg tissue, but following 20 min of low-flow ischaemia (0.56 ml.min$^{-1}$.gww$^{-1}$), it accumulated to 23.9 nmol per mg tissue (p<0.01) before returning to 0.7 nmol per mg tissue following reperfusion. β-OHB was also seen to rise in the coronary effluent throughout ischaemia, returning to a lower level following reperfusion, suggesting that the myocardial β-OHB was oxidised by the heart upon reperfusion. The myocardial accumulation and efflux of β-OHB during ischaemia thereby mirrored that of lactate (*Figure 2C–D*). In this protocol, cardiac contractile function recovered to 100% of its pre-ischaemic level upon reperfusion (*Figure 2—figure supplement 1*).

The ischaemic accumulation of β-OHB was associated with a 2.6-fold increase in myocardial acetyl-CoA during ischaemia (*Figure 2E*; p<0.05), a 3.6-fold increase in acetoacetyl-CoA (*Figure 2F*; p<0.0001), and a 3.4-fold increase in HMG-CoA relative to the pre-ischaemic LV (*Figure 2G*; p<0.01). The unexpected finding of increased HMG-CoA suggested that some flux may occur through HMGCS2 in the ischaemic heart, despite its low activity in the heart in comparison with liver (*McGarry and Foster, 1976*). The lipid peroxidation marker malondialdehyde (MDA) was also 2.3-fold higher in the ischaemic heart compared with the non-ischaemic heart, demonstrating oxidative stress (*Figure 2—figure supplement 2*; p<0.05).

### Ischaemia and Krebs cycle intermediates

Next, we investigated the impact of ischaemia upon the levels of Krebs cycle intermediates. As expected, 4-carbon intermediates accumulated during ischaemia (*Figure 3*), including a 4.3-fold increase in succinate (p<0.01), which is known to augment the reduction of acetoacetate to β-OHB (*Schönfeld et al., 2010*). Meanwhile, myocardial citrate concentrations during ischaemia were 65% lower than pre-ischaemic levels (p<0.05), which in conjunction with the accumulation of acetyl-CoA (*Figure 2E*) suggested lower citrate synthase activity during ischaemia.

### Lactate dehydrogenase inhibition enhanced β-hydroxybutyrate accumulation during ischaemia

We next sought to enhance the production of glucose-derived acetyl-CoA during ischaemia to investigate whether this influenced β-OHB production. To do this, hearts were perfused with sodium oxamate, a competitive lactate dehydrogenase (LDH) inhibitor. As expected, when oxamate was present, lactate efflux into the coronary effluent was 4.4-fold lower over the course of the ischaemic period in comparison with non-treated hearts (*Figure 4A*). LV lactate levels were unexpectedly elevated 1.44-fold during ischaemia as a result of oxamate (*Figure 4B*; p<0.01), perhaps suggesting impaired lactate efflux. In accordance with the expected effects of LDH inhibition, pyruvate and acetyl-CoA concentrations were 2.5- and 1.5-fold higher, respectively, following oxamate treatment (*Figure 4C–D*; p<0.01 and p<0.05, respectively).

The increase in ischaemic acetyl-CoA was reflected in 7.2-fold greater levels of acetoacetyl-CoA and 1.8-fold greater levels of HMG-CoA (*Figure 4E–F*; p<0.0001 and p<0.05). Sodium oxamate treatment also resulted in 5.3-fold greater β-OHB accumulation during ischaemia (*Figure 4G*; p<0.001). There was also a 2.7-fold greater initial secretion of β-OHB into the perfusate during ischaemia at 36 min (*Figure 4—figure supplement 1*; p<0.001), although this secretion decreased as the ischaemic period progressed, reaching 0 at 60 min before again rising to 44-fold higher than pre-ischaemia upon reperfusion (p<0.0001). Furthermore, succinyl-CoA was present at 2.2-fold greater levels in oxamate-treated hearts during ischaemia compared with untreated hearts, whilst succinate levels were 32% lower (*Figure 4H–I*; p<0.001 and p<0.0001), suggesting that reverse flux

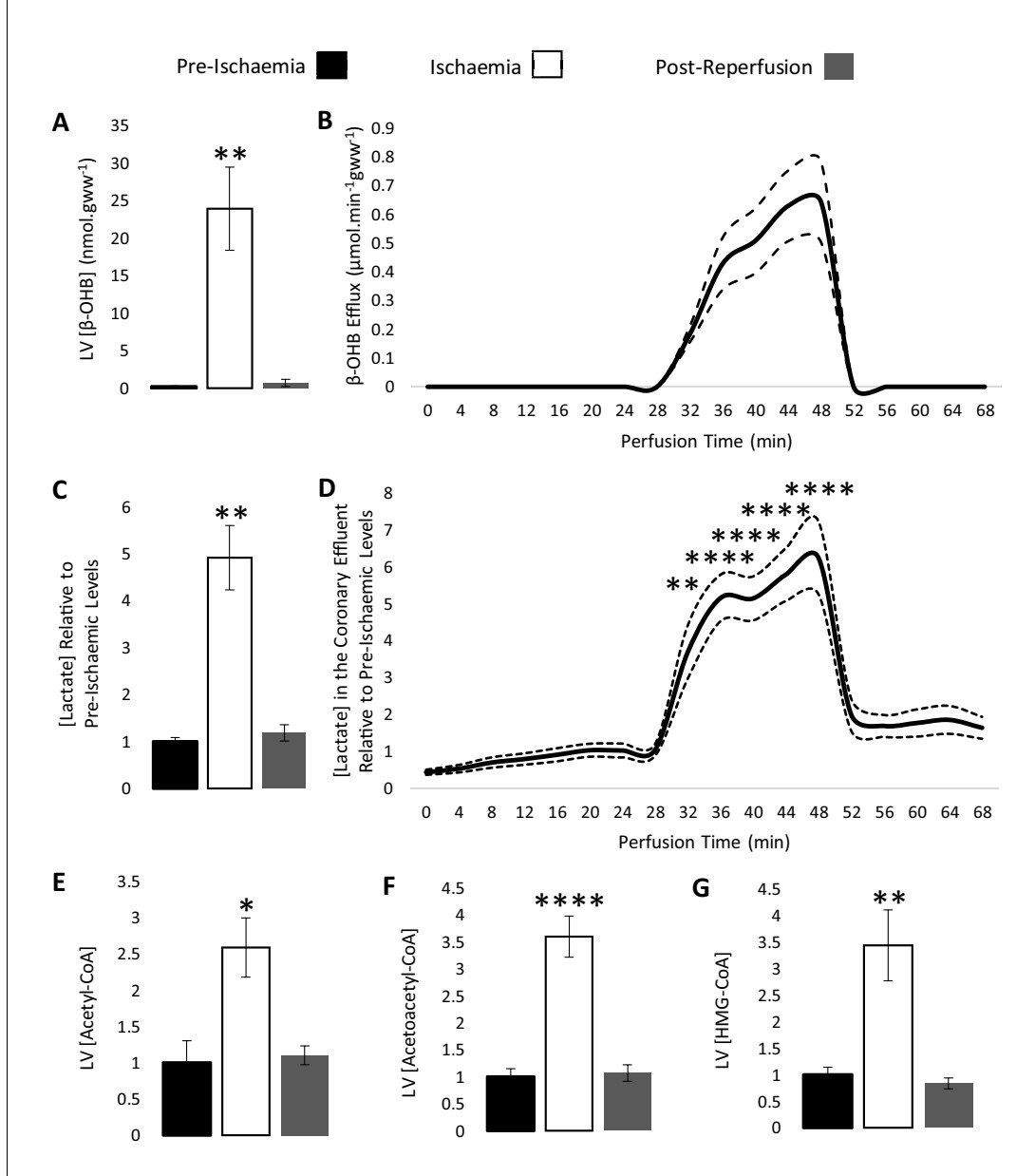

**Figure 2.** β-hydroxybutyrate accumulates in the ischaemic rat heart. Concentrations of β-hydroxybutyrate (β-OHB) in rat heart and the coronary effluent relative to pre-ischaemic levels (**A** and **B**), and those of lactate (**C** and **D**). Pre-ischaemic, ischaemic, and post-reperfusion concentrations of acetyl-coenzyme A (acetyl-CoA), acetoacetyl-CoA, and β-hydroxy-β-methylglutaryl-CoA (HMG-CoA) in the heart relative to pre-ischaemic levels (**E**, **F**, and **G**). All three groups contained n = 7 hearts. Results are displayed as mean ± SEM. *p<0.05, **p<0.01, ***p<0.001, and ****p<0.0001, relative to pre-ischaemic levels.

The online version of this article includes the following figure supplement(s) for figure 2:

**Figure supplement 1.** Cardiac contractile recovery of reperfused hearts for *Figure 2*.

**Figure supplement 2.** Pre-ischaemic, post-ischaemic, and post-reperfusion LV levels of the lipid peroxidation marker malondialdehyde (MDA), relative to pre-ischaemic levels (three groups of n = 7).

**Figure supplement 3.** Ischaemic β-OHB levels with and without Intralipid in the perfusion buffer.

through SCOT contributes to the production of β-OHB in the ischaemic heart (*Figure 4J*). Sodium oxamate administration, and its associated metabolic changes, led to slower recovery of contractile function post-reperfusion in comparison with control hearts (*Figure 4—figure supplement 2*).

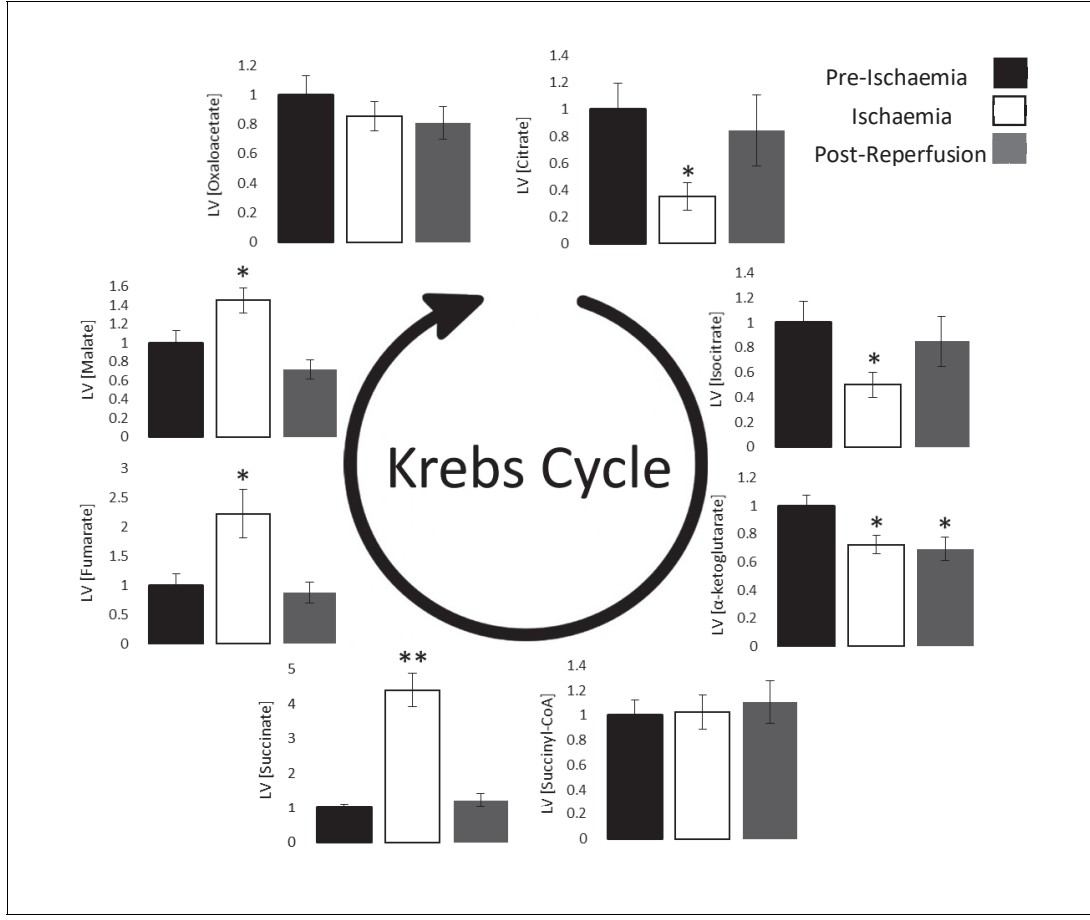

**Figure 3.** Ischaemic Krebs cycle flux. Concentrations of Krebs cycle intermediates in the left ventricle pre-ischaemia, at the end of the ischaemic period and post-reperfusion, all relative to pre-ischaemic levels. All three groups comprised n = 7 hearts. Results are displayed as mean ± SEM. *p<0.05, **p<0.01, relative to pre-ischaemic levels.

Ischaemic levels of lipid peroxidation were also 51% greater than in control hearts (*Figure 4—figure supplement 3*; p<0.05).

## Inhibition of HMGCS2 attenuated β-hydroxybutyrate accumulation during ischaemia

To investigate possible flux through HMGCS2 during ischaemia, hearts were perfused with the HMG-CoA synthase inhibitor hymeglusin. Treatment with hymeglusin resulted in 27% lower ischaemic lactate accumulation (*Figure 5A*; p<0.05), while pyruvate concentrations in ischaemia were unaffected (*Figure 5B*). Meanwhile, acetyl-CoA levels were 5.1-fold greater (*Figure 5C*; p<0.01) in hearts where HMGCS2 was inhibited. Hymeglusin administration led to a 7.6-fold ischaemic accumulation of acetoacetyl-CoA (a substrate for HMGCS2), and a 65% lower level of HMG-CoA (*Figure 5D–E*; both p<0.01), suggesting that inhibition of HMGCS2 was effective. This also resulted in a 40% lower ischaemic accumulation of β-OHB in the LV compared with vehicle-treated hearts (*Figure 5F*; p<0.01). Moreover, hymeglusin resulted in a 41% lower level of succinate and a 2.7-fold greater level of succinyl-CoA than in non-treated hearts (*Figure 5G–H*; both p<0.05), which is consistent with the accumulation of acetoacetyl-CoA driving greater reverse flux through SCOT (*Figure 5I*).

## HMGCS2 inhibition was cardioprotective and conserved mitochondrial capacity

Hearts were subjected to a more exacting 32-min 0.3-ml.min$^{-1}$gww$^{-1}$ ischaemic period in order to assess whether hymeglusin influenced functional recovery following reperfusion for 32 min.

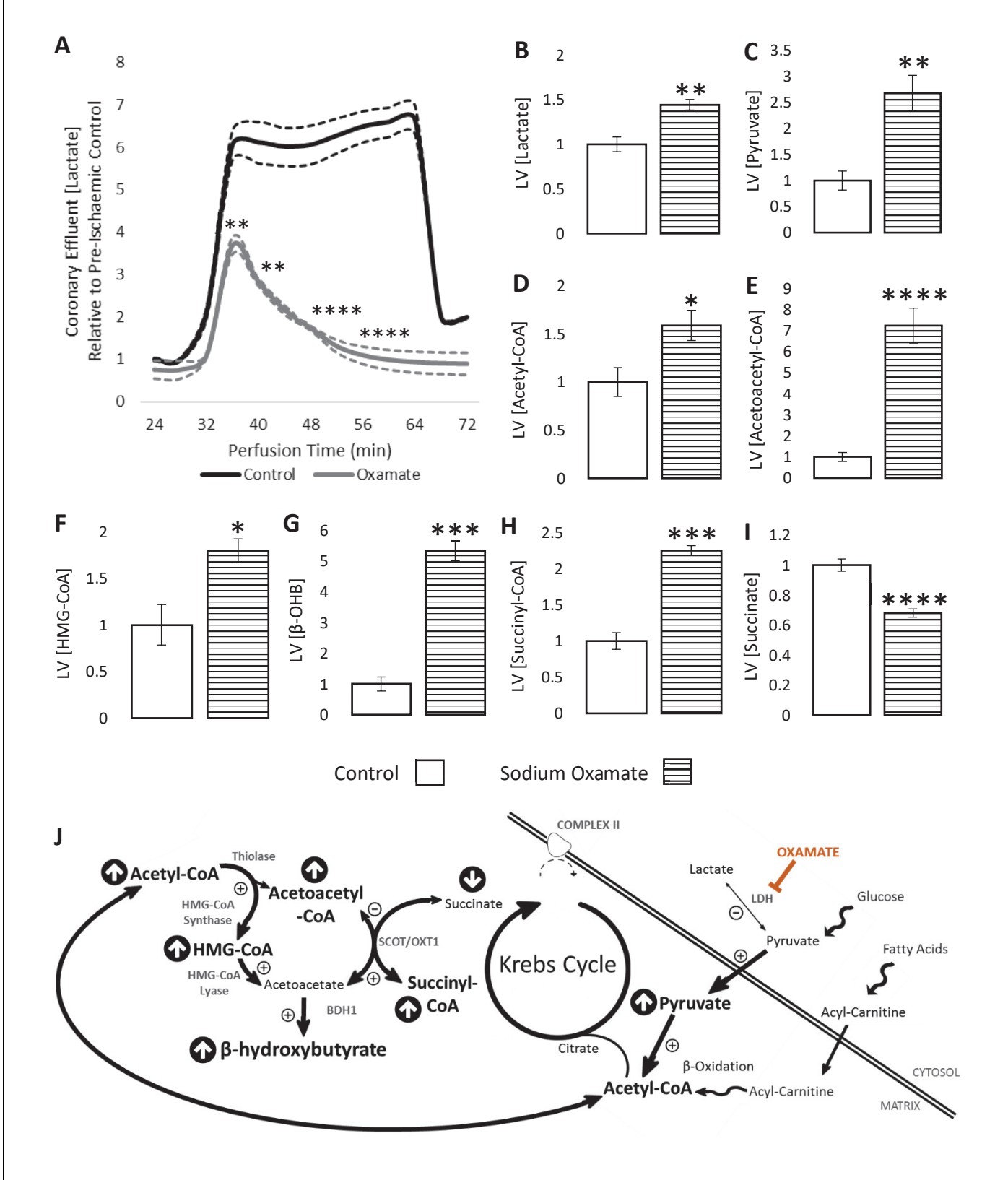

**Figure 4.** LDH inhibition enhances ketone flux in ischaemia. (**A**) Coronary effluent lactate levels relative to control pre-ischaemic levels, with and without delivery of 50 mM of the lactate dehydrogenase (LDH) inhibitor sodium oxamate in the perfusion buffer (two groups of n = 5 hearts). Relative to

*Figure 4 continued on next page*

*Figure 4 continued*

ischaemic controls, the ischaemic LV concentrations of (B) lactate, (C) pyruvate, (D) acetyl-coenzyme A (acetyl-CoA), (E) acetoacetyl-CoA, (F) β-hydroxy-β-methylglutaryl-CoA (HMG-CoA), (G) β-hydroxybutyrate (β-OHB), (H) succinyl-CoA, and (I) succinate with and without delivery of 50 mM of the LDH inhibitor sodium oxamate in the perfusion buffer (two groups, n = 6 hearts each). (J) Schematic depicting the suggested diversion of pyruvate away from lactate production and into β-OHB synthesis. Bold text and upward pointing arrows indicate metabolites which accumulated to a greater level in the presence of oxamate, downward pointing arrows and smaller text indicate those which accumulated to a lesser extent. Bold arrows suggest pathways where flux may be increased in the presence of oxamate. Results are presented as mean ± SEM. *p<0.05, **p<0.01, ***p<0.001, and ****p<0.0001, relative to control.

The online version of this article includes the following figure supplement(s) for figure 4:

**Figure supplement 1.** Coronary effluent β-OHB levels relative to control pre-ischaemic levels, with and without delivery of 50 mM sodium oxamate in the perfusion buffer (two groups of n = 5 hearts each).

**Figure supplement 2.** Inhibition of lactate dehydrogenase delays functional recovery.

**Figure supplement 3.** The ischaemic LV MDA concentration with and without treatment with 50 mM of the LDH inhibitor sodium oxamate in the perfusion buffer, relative to ischaemic controls (two groups of n = 6 hearts each).

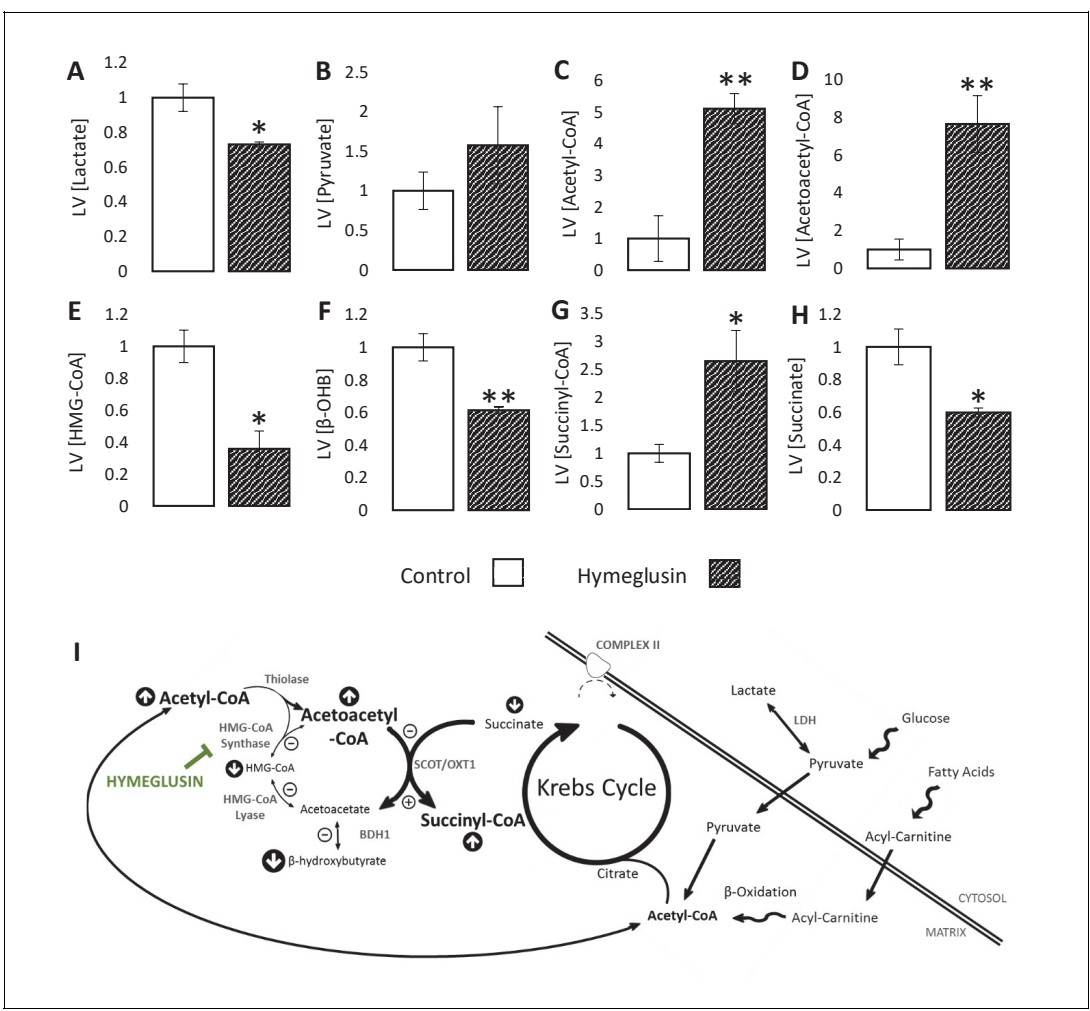

**Figure 5.** Inhibition of the ketogenic enzyme HMG-CoA synthase impairs ketone flux. Relative to ischaemic controls, the ischaemic LV concentrations of (A) lactate, (B) pyruvate, (C) acetyl-coenzyme A (acetyl-CoA), (D) acetoacetyl-CoA, (E) β-hydroxy-β-methylglutaryl-CoA (HMG-CoA), (F) β-hydroxybutyrate, (G) succinyl-CoA, and (H) succinate, with and without delivery of 2.5 µM of the HMG-CoA synthase (HMGCS) inhibitor hymeglusin in the perfusion buffer (two groups, n = 6 hearts each). (I) Schematic depicting how the inhibition of HMGCS is associated with metabolite level changes. Bold text and upward pointing arrows indicate metabolites that accumulated to a greater level in the presence of hymeglusin, downward pointing arrows and smaller text indicate those that accumulated to a lesser extent. Bold arrows suggest pathways where flux may be increased in the presence of hymeglusin. Results are presented as mean ± SEM. *p<0.05, **p<0.01, relative to ischaemic control.

Hymeglusin administration 12 min before ischaemia resulted in 33% greater recovery of cardiac contractile function upon reperfusion, relative to untreated controls (*Figure 6A*). At 52, 56, and 60 min during the ischaemic period, 40, 43, and 35% lower levels of β-OHB were detected in the coronary effluent from hymeglusin-administered hearts relative to control hearts (*Figure 6—figure supplement 1*; $p<0.01$, $p<0.05$, and $p<0.05$, respectively). Moreover, ischaemia/reperfusion (I/R) resulted in impaired mitochondrial respiratory capacities relative to control hearts perfused for 96 min without ischaemia (*Figure 6B*), with 36% lower β-oxidation-supported

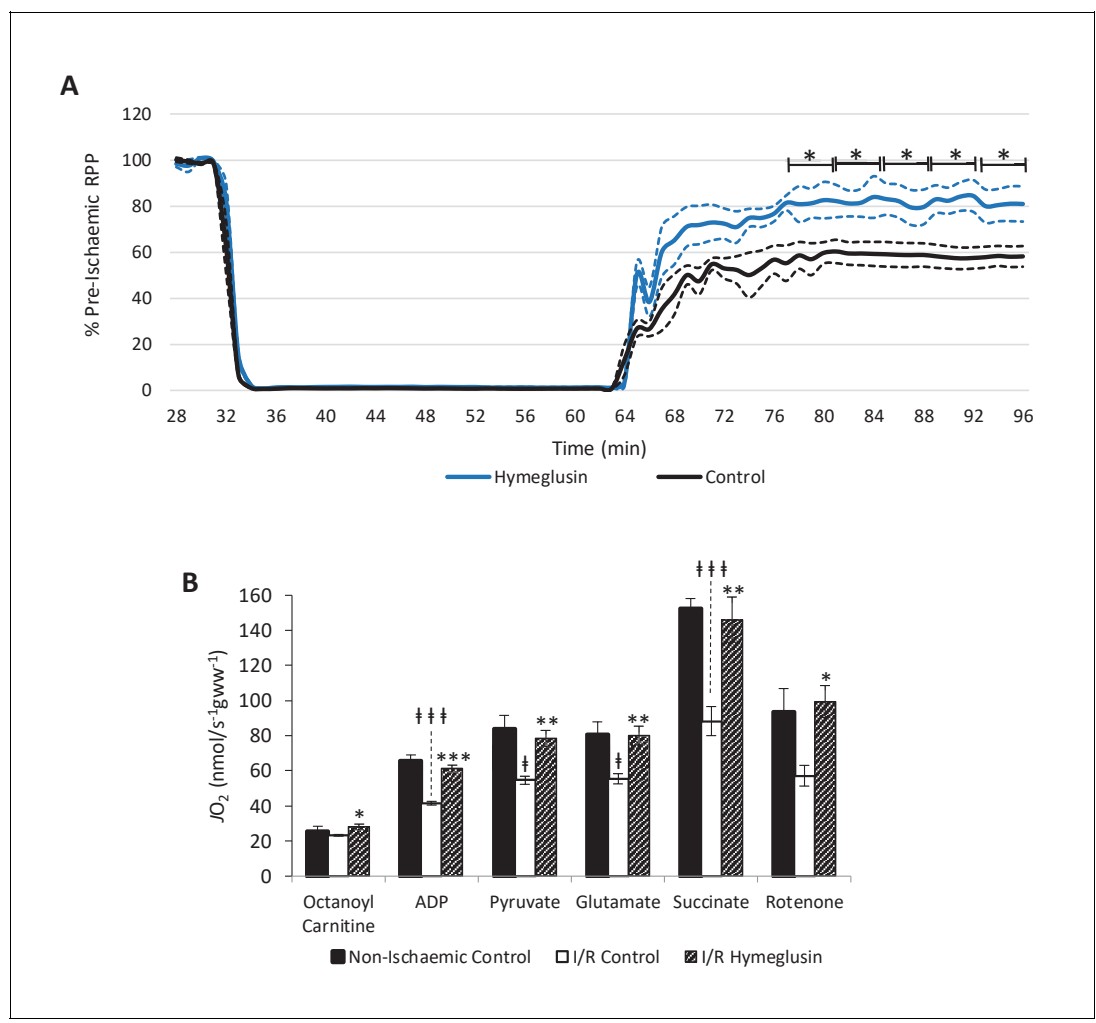

**Figure 6.** Inhibition of HMG-CoA synthase improves functional recovery. (**A**) Rate pressure product (RPP), as a percentage of pre-ischaemic levels, following 32 min of 0.3 ml.min$^{-1}$gww$^{-1}$ low-flow ischaemia, with or without delivery of 2.5 µM of the β-hydroxy-β-methylglutaryl (HMG)-CoA synthase (HMGCS) inhibitor hymeglusin in the perfusion buffer (two groups, n = 5 hearts each). (**B**) Respiration rates corrected for wet mass. Malate and octanoyl carnitine were added initially to stimulate leak respiration, then, sequentially, adenosine diphosphate (ADP) to stimulate β-oxidation-supported OXPHOS, pyruvate, and glutamate to support electron flux through complex I, succinate to additionally support electron flux through complex II, and finally rotenone to inhibit complex I- and isolate complex II-supported respiration. The non-ischaemic control group consisted of n = 5 hearts perfused aerobically for 96 min. The I/R control and I/R hymeglusin groups consisted of n = 5 hearts per group subjected to 32 min of aerobic perfusion at 100 mmHg, 32 min of 0.3 ml.min$^{-1}$gww$^{-1}$ low-flow ischaemia, then 32 min of aerobic reperfusion at 100 mmHg, with either vehicle or 2.5 µM hymeglusin administered in the perfusion buffer. Results are presented as mean ± SEM. *$p<0.05$, **$p<0.01$, relative to ischaemic control. †$p<0.05$, #$p<0.01$, and ‡‡‡$p<0.001$, relative to non-ischaemic control.

The online version of this article includes the following figure supplement(s) for figure 6:

**Figure supplement 1.** Coronary effluent β-OHB levels relative to control pre-ischaemic levels, with and without delivery of hymeglusin in the perfusion buffer (two groups of n = 5 hearts).

**Figure supplement 2.** The ischaemic LV MDA concentration with and without treatment with 2.5 µM of the HMG-CoA synthase (HMGCS) inhibitor hymeglusin in the perfusion buffer, relative to ischaemic controls (two groups of n = 6 hearts each).

oxidative phosphorylation (OXPHOS) capacity (p<0.001), 36% lower β-oxidation + pyruvate-supported OXPHOS capacity (p<0.05), 31% lower complex I capacity (p<0.05), and a 42% impairment of respiratory capacity supported by complex I and II (p<0.001). Administration of hymeglusin prior to I/R attenuated the post-ischaemic impairment of cardiac mitochondrial respiratory capacity, with no significant difference between respiratory capacity of hymeglusin-treated I/R hearts and those of time-matched non-ischaemic controls. Complex II capacity was not significantly affected by ischaemia; however, hymeglusin resulted in a post-reperfusion complex II capacity which was 74% greater than in the vehicle-treated heart (p<0.05) and not significantly different to non-ischaemic time-matched controls. Hymeglusin administration also resulted in 38% lower ischaemic levels of the lipid peroxidation marker malondialdehyde (*Figure 6—figure supplement 2*; p<0.05).

## Discussion

Here we sought to test the hypothesis that β-OHB accumulates in the Langendorff-perfused rat heart under conditions of low-flow ischaemia, in association with elevated levels of acetyl-CoA and suppressed Krebs cycle activity. We found that β-OHB accumulated to 23.9 nmol per mg myocardial tissue and also appeared in the coronary effluent. Following reperfusion, β-OHB was absent from the perfusate and was greatly lowered in the myocardium, probably as a result of oxidation. This production of β-OHB was enhanced following inhibition of LDH. Whilst most myocardial β-OHB appeared to result from reverse flux through SCOT, we unexpectedly found that at least some β-OHB production occurred via the production of HMG-CoA by HMGCS2. Indeed, inhibition of HMGCS2 lowered myocardial levels of HMG-CoA and suppressed β-OHB in the ischaemic heart despite an apparent increase in reverse flux through SCOT. This was associated with enhanced functional recovery from ischaemia/reperfusion and protection of mitochondrial respiratory function in the post-ischaemic rat heart.

Our approach exploited the combination of mass spectrometry and Langendorff perfusion, which allowed a tight control of metabolic substrate supply to the heart, which is unattainable in vivo. Mass spectrometry allowed us to detect β-OHB and neighbouring metabolites, under aerobically perfused and ischaemic conditions. Use of the LDH inhibitor sodium oxamate allowed us to investigate the implications of enhanced glucose-derived acetyl-CoA upon candidate pathways. We do, however, urge caution when interpreting some of the results of our experiments with LDH inhibition, particularly regarding the functional recovery from I/R, since this is likely to be more substantially influenced by impaired lactate production than by changes in β-OHB or associated metabolites (*Griffin et al., 2000*). Moreover, despite the high concentration of sodium oxamate used here being in line with that used by others previously (*Yoshioka et al., 2012*), it is such that we cannot exclude osmotic effects. The use of hymeglusin allowed us to probe the unexpected finding that flux through HMGCS2 during ischaemia supports the production of HMG-CoA and consequently of β-OHB. We cannot definitively exclude off-target effects of hymeglusin, though we are not aware of any at the concentration used here. Moreover, our data showing accumulation of acetoacetyl-CoA coupled with depletion of HMG-CoA indicates that the pathway was inhibited. Hymeglusin treatment itself appeared to increase the reverse flux through SCOT during ischaemia, as evidenced by decreased succinate and increased succinyl-CoA levels relative to untreated hearts. Whilst our results therefore suggest that both SCOT- and HMGCS2-supported pathways contribute to the ischaemic accumulation of β-OHB, they do not indicate the relative contributions of each, and flux through SCOT may be the more significant route.

Our findings add to those of others who have previously reported ketone body production by the ex vivo perfused heart (*Comte et al., 1997*; *Opie and Owen, 1975*), in myocardial extracts (*Stern et al., 1953*) and in isolated cardiac mitochondria (*LaNoue et al., 1970*), and support the notion that ketone body production may represent a spillover pathway for excess acetyl-CoA. Whilst it has not been conclusively demonstrated, a number of studies have suggested that myocardial ketone body production may also occur under hypoxic/ischaemic conditions in vivo. In studies of dogs, hypoxaemia resulted in enhanced β-OHB:acetoacetate and net myocardial acetoacetate production in some animals, whilst this was not seen in any non-hypoxaemic dogs (*Whereat and Chan, 1972*). Notably, those animals exhibiting net acetoacetate production had arterial lactate concentrations in excess of 5 mM, suggesting a high degree of hypoxic stress (*Whereat and Chan, 1972*). The same investigators also reported net cardiac acetoacetate production in one dog following

myocardial infarction (*Whereat and Chan, 1972*). More recently, it has been reported that serum levels of β-OHB and acetoacetate were found to increase systemically in human patients following balloon-induced coronary ischaemia during elective coronary angioplasty, which was suggested to reflect the early phase of metabolic adjustment during ischaemia (*Di Marino et al., 2018*). Meanwhile, myocardial β-OHB utilisation was suppressed during myocardial ischaemia in patients experiencing angina-like chest pain (*Arima et al., 2019*).

Efflux of β-OHB is seen shortly after the onset of ischaemia, and its production is therefore likely to result from altered substrate availability and/or post-translational modifications rather than altered gene expression. Acetyl-CoA and succinate accumulate during ischaemia, while mitochondrial reactive oxygen species (ROS) are also known to be produced (*Chen et al., 2003*). Each of these intermediates may post-translationally modify proteins, and thereby support ketone flux. As such, this could be an interesting avenue to further elucidate mechanisms.

An unexpected finding of our study was that at least some of the β-OHB produced in the ischaemic heart occurred via HMGCS2. HMGCS2 is expressed in human heart, albeit at a lower level than in liver (*Mascaró et al., 1995*), and whilst activity levels of HMGCS2 and HMG-CoA lyase are, respectively, 20-fold and 6.8-fold higher in adult rat liver than heart, activity in the myocardium exceeds that in brain or skeletal muscle (*McGarry and Foster, 1976*). Although the role of cardiac HMGCS2 is unknown, it is transiently expressed at a high level in neonatal mouse heart before declining by postnatal day 23 (*Talman et al., 2018*) and has been shown to be upregulated in type 1 diabetes following streptozotocin (STZ) treatment in rats (*Cook et al., 2017*) and mice (*Shukla et al., 2017*), as well as in mice when fasted or on a ketogenic diet (*Wentz et al., 2010*). Meanwhile, a recent preliminary report suggested that heart-specific overexpression of HMGCS2 resulted in increased cardiac β-OHB, along with mitochondrial swelling and systolic impairment (*Zenimaru et al., 2018*).

In line with this latter report, we found that inhibition of HMGCS2 was associated with protection of mitochondrial respiratory capacity in the post-ischaemic rat heart and enhanced functional recovery from ischaemia/reperfusion. This protection is unlikely to arise as a direct result of decreased β-OHB availability; ketone body administration has been shown to be neuroprotective during ischaemic stroke injury (*Yin et al., 2015*) and during renal ischaemia/reperfusion (*Tajima et al., 2019*). Whilst the basis for this protection remains unclear, our findings do hint at a number of possible mechanisms. The increased availability of acetyl-CoA may be protective in supporting Krebs cycle activity and oxidative phosphorylation upon reperfusion. Alternatively, the inhibition of HMGCS2 supported greater reverse flux through SCOT, thereby lowering the ischaemic accumulation of succinate. Succinate accumulation during ischaemia is associated with ROS production via reverse electron transfer (RET) to complex I as the succinate becomes oxidised upon reperfusion (*Chouchani et al., 2014*; *Schönfeld et al., 2010*). It has also been demonstrated previously that reduction of this ischaemic succinate accumulation via dimethyl malonate administration protects against I/R injury (*Chouchani et al., 2014*). Inhibition of HMGCS2 may therefore represent a novel cardioprotective strategy, perhaps of particular relevance to the diabetic heart, where sensitivity to ischaemia/reperfusion injury (*Paulson, 1997*) and, at least in the case of type 1 diabetes, expression of HMGCS2 are both increased (*Cook et al., 2017*; *Shukla et al., 2017*).

## Conclusions

Ketone body accumulation occurs in the heart under conditions of low-flow ischaemia, and this is mediated at least in part by HMG-CoA synthase. Inhibition of HMG-CoA synthase by hymeglusin reduces the accumulation of β-OHB during ischaemia, and protects both cardiac contractile function and mitochondrial respiratory capacity against damage mediated by I/R. This finding sheds new light on the role of HMG-CoA synthase in the heart, and provides context for studies reporting altered cardiac ketone body metabolism under pathological conditions.

## Materials and methods

Key resources table

*Continued on next page*

*Continued*

| Reagent type (species) or resource | Designation | Source or reference | Identifiers | Additional information |
|---|---|---|---|---|
| Reagent type (species) or resource | Designation | Source or reference | Identifiers | Additional information |
| Strain, strain background (*Rattus Norvegicus*) | Wistar Rat | Charles River | Strain Code 003; RRID:RGD_13508588 | 300–350 g |
| Chemical compound, drug | Hymeglusin | Sigma Aldrich | SML0301 | |
| Chemical compound, drug | Sodium Oxamate | Sigma Aldrich | O2751 | |

## Animal studies

### Ethical approval

All experiments conformed to UK Home Office guidelines under the Animals in Scientific Procedures Act and were approved by the University of Cambridge Animal Welfare and Ethical Review Committee.

Power calculations were performed using a 'known' mean of 1, an 'expected' mean of 0.8, and a sigma of 0.18 for a significance of 0.05, generating a suggested sample size of 7 for the initial experiments. This is consistent with the power and group sizes with which significance has previously been achieved in similar studies.

## Materials and reagents

All reagents were obtained from Sigma Aldrich unless otherwise stated.

## Heart perfusion

Male Wistar rats (300–350 g; n = 73) were obtained from a commercial breeder (Charles River, Margate, UK) and housed in conventional cages with a normal 12-hr/12-hr light/dark photoperiod and access to normal rodent chow and water ad libitum. Rats were euthanised by rising $CO_2$ levels, with death confirmed by cervical dislocation. Hearts were excised and perfused in the Langendorff mode with Krebs-Henseleit (KH) buffer (118 mM NaCl, 4.7 mM KCl, 1.2 mM $MgSO_4$, 1.3 mM $CaCl_2$, 0.5 mM ethylenediaminetetraacetic acid (EDTA), 25 mM $NaHCO_3$, 1.2 mM $KH_2PO_4$; pH 7.4) plus substrates as detailed below, gassed with 95% $O_2$-5% $CO_2$. Heart temperature was monitored throughout and maintained at 38°C. Functional parameters were measured using a PVC balloon inserted into the left ventricle; rate-pressure product (RPP) was calculated as the left ventricular developed pressure (LVDP) x heart rate. Hearts were randomly assigned into the experimental groups for each experiment. A 1 ml coronary effluent sample was obtained every 4 min throughout each protocol and frozen for metabolic analysis. Two different perfusion protocols were used, which differed in the degree of ischaemic insult.

First we used a relatively mild ischaemic protocol to investigate ketone metabolism under conditions which did not induce overt functional impairment upon reperfusion. The perfusate was KH buffer containing 11 mM glucose and 0.4 mM equivalent triglyceride in the form of Intralipid. The perfusion protocol entailed a 32-min aerobic perfusion followed by 20 min of low-flow ischaemia (0.56 ml.min$^{-1}$.gww$^{-1}$) and 20 min of reperfusion. The protocol was terminated by snap-freezing the left ventricle either at the end of the aerobic period, at the end of the ischaemic period before reperfusion or following reperfusion (n = 7 for each group).

Second, to investigate the impact of altering ketone flux on functional recovery from ischaemia/reperfusion, we used a perfusion protocol with a more severe ischaemic challenge. Here, the perfusate comprised KH buffer with 11 mM glucose alone, since we observed no difference in ischaemic β-OHB accumulation in hearts perfused with glucose and Intralipid compared with those perfused with glucose alone (*Figure 2—figure supplement 3*). Following 32 min of aerobic perfusion at a constant

pressure of 100 mmHg, hearts were subjected to 32 min of low-flow ischaemia at 0.32 ml.min$^{-1}$. gww$^{-1}$, followed by 32 min of reperfusion. To manipulate possible pathways driving ketone flux, either sodium oxamate (50 mM), a competitive inhibitor of LDH (*Yoshioka et al., 2012*), or hymeglusin (2.5 µM), an inhibitor of HMG-CoA synthase (*Le Foll et al., 2014*; *Skaff et al., 2012*), or vehicle (KH for oxamate or 80 µl dimethyl sulfoxide [DMSO] for hymeglusin) was added to the buffer 12 min before induction of ischaemia (n = 5 for each group). Hymeglusin was delivered at a concentration which exceeded that demonstrated to inhibit HMGCS2 in cardiomyocytes (*Talman et al., 2018*), whilst oxamate was administered at a concentration previously used in the ex vivo perfused mouse heart (*Yoshioka et al., 2012*).

Alongside the hymeglusin- and vehicle-administered hearts, a third group of hearts (n = 5) was perfused aerobically for 96 min without ischaemia/reperfusion, as a non-ischaemic time-matched control for mitochondrial respiratory analysis. From each of these three groups, a section of non-frozen left ventricle (~5 mg) was placed in ice-cold BIOPS solution (2.77 mM CaK$_2$-ethylene glycol tetraacetic acid [EGTA], 7.23 mM K$_2$EGTA, 6.56 mM MgCl$_2$.6H$_2$O, 20 mM taurine, 15 mM phosphocreatine, 20 mM imidazole, 0.5 mM dithiothreitol, 50 mM 2-(*N*-morpholino)ethanesulfonic acid [MES], 5.77 mM Na$_2$ATP; pH 7.1) for immediate analysis of mitochondrial respiratory capacity using high-resolution respirometry.

Finally, a further group of hearts perfused with hymeglusin, sodium oxamate, or vehicle (n = 6 per condition) were snap-frozen at the end of the ischaemic period without reperfusion, for the assessment of myocardial metabolite levels. Metabolite levels were not assessed pre-ischaemia or post-reperfusion in these experiments owing to the lack of any β-OHB seen pre-ischaemia in the initial experiments (*Figure 2*).

Absolute cardiac function was consistent between all groups. For hearts where the functional recovery is represented as a percentage of pre-ischaemic contractile function, absolute functional values are detailed in *Supplementary files 1–3*.

## Metabolite extraction from tissue

Metabolites were extracted from frozen LV samples using a methanol/chloroform/water extraction method (*Le Belle et al., 2002*). Frozen tissues were added to 600 µL methanol/chloroform (2:1; v/v), and the samples homogenised with a Tissuelyser (Qiagen, UK) for 5 min at a frequency of 20/s before 15 min of sonication. Water (200 µL) and chloroform (200 µL) were then added to the samples prior to centrifugation at 17,000 x*g* for 7 min. The resulting aqueous and organic phases were collected and the extraction procedure repeated for a second time on the protein pellets. Aqueous phases were dried down using an evacuated centrifuge, while the organic phase was dried by evaporation under a stream of nitrogen. Dried samples were stored at −20°C until further analysis.

### BEH amide LC methodology

Aqueous extract fractions were reconstituted in an acetonitrile: 10 mM ammonium carbonate solution (7:3 v/v, 100 µl) containing a 10 µM mixture of internal standards (200 µL; phenylalanine d5, valine d8, leucine d10). The column used was a 1.7-µm BEH amide column (150 x 2.1 mm), coupled to a Vanquish UHPLC+ series (Thermo Scientific, UK) LC system and a TSQ Quantiva Triple Quadrupole Mass Spectrometer (Thermo Scientific). The mobile phase was pumped at 600 µl min$^{-1}$ with 0.1% ammonium carbonate solution as mobile phase A and acetonitrile as mobile phase B. Mobile phase A was held at 20% for 1.5 min, linearly increased to 60% over the next 2.5 min, and held at 60% for 1 min, before being decreased back to the initial conditions (20% mobile phase A) in the next 0.1 min. The total run time was 6 min. Nitrogen at 48 mTorr, 420°C, was used as a drying gas for solvent evaporation and the ultra performance liquid chromatography column was conditioned at 30°C.

### C18pfp LC methodology

Heart sample extracts were reconstituted in a 10 mM ammonium acetate solution/internal standard mixture (200 µL; phenylalanine d5, valine d8, leucine d10), while coronary effluent samples were diluted (20 µl plus 80 µl 10 mM ammonium acetate/internal standard mix). All samples were run for 6 min using an ACE Excel-2 C18-PFP 5 µm column (100 A, 150 x 2.1 mm, 30°C) on either a Thermo Vanquish LC system coupled to a Thermo Quantiva triple quadrupole mass spectrometer for

targeted analysis or a Thermo Dionex Ultimate 3000 LC system coupled to a Thermo Elite orbitrap mass spectrometer for open profiling. Mobile phase A was 0.1% formic acid, while mobile phase B was acetonitrile plus 0.1% formic acid. The LC gradient was as follows: 0% B for 1.6 min followed by a linear gradient up to 30% B for 2.4 min. There was a further linear increase to 90% B for 30 s, following which B was held at 90% for 30 s before re-equilibration for 1.5 min. Drying gas was as used in the BEH amide method.

### Detection

The Thermo Elite orbitrap mass spectrometer had run parameters as follows: heater temperature 420°C, sheath gas flow rate 60 units, aux gas flow rate 20 units, and sweep gas flow rate 5 units. The spray voltage was −2.5, capillary temperature 380°C, and S-lens RF level 60%. On the Quantiva, electrospray voltage switched rapidly between 3.5 kV and −2.5 kV.

### β-hydroxybutyrate quantification

To allow quantification of β-hydroxybutyrate, a standard dilution series comprising eight known concentrations of β-hydroxybutyrate between 100 µM and 50 nM was run in tandem with the biological samples using the Thermo Elite orbitrap. β-hydroxybutyrate was measured at m/z = 103.025, with a retention time of 2.2 min.

## High-resolution respirometry

A section of the left ventricle 5 mm above the apex was dissected into bundles of six to eight myofibres and permeabilised with gentle rocking for 20 min at 4°C in BIOPS with 50 µg.ml$^{-1}$ saponin (*Pesta and Gnaiger, 2012*). Fibre bundles were then washed (3 x 5 min, 4°C) in respiration medium (MiR05: 0.5 mM EGTA, 3 mM MgCl$_2$.6H$_2$O, 60 mM K-lactobionate, 20 mM taurine, 10 mM KH$_2$PO$_4$, 20 mM (4- (2-hydroxyethyl) -1-piperazineethanesulfonic acid [HEPES]), 110 mM sucrose, 1 g L$^{-1}$ defatted bovine serum albumin [BSA], pH 7.1).

Approximately 2 mg of the resulting fibre bundles was immediately added to Oxygraph-O2k chambers (Oroboros Instruments, Innsbruck, Austria) containing 2 ml MiR05 at 37°C. Respiratory capacities were assessed as described previously (*Horscroft et al., 2015*). Briefly, malate (2 mM) and octanoyl carnitine (0.2 mM) were added initially to stimulate LEAK respiration, followed by adenosine diphosphate (ADP) (5 mM) to stimulate β-oxidation-supported OXPHOS, then pyruvate (20 mM) for a comparison of substrate preference, glutamate (10 mM) to support the N-pathway via complex I, cytochrome *c* (10 µM) to assess outer mitochondrial membrane integrity, succinate (10 mM) to activate the S-pathway via complex II, and rotenone (0.5 µM) to inhibit complex I.

## Statistical analysis

Measured values were compared to those in pre-ischaemic or non-ischaemic control groups as appropriate using a two-tailed Student's t-test. To assess changes in functional recovery, the average RPP across 4-min sections was compared to the same 4-min section in control hearts using a two-tailed Student's t-test. A p-value less than or equal to 0.05 was considered statistically significant.

## Supporting data

Data from this study is available at the University of Cambridge Online Data Repository https://doi.org/10.17863/CAM.72871.

---

## Additional information

### Competing interests

Ross T Lindsay: Ross T. Lindsay is affiliated with BioPharmaceuticals R&D, AstraZeneca Ltd. The author has no financial interests to declare. The other authors declare that no competing interests exist.

---

## Funding

| Funder | Grant reference number | Author |
| --- | --- | --- |
| British Heart Foundation | FS/14/59/31282 | Ross T Lindsay |
| Research Councils UK | EP/E500552/1 | Andrew Murray |

The funders had no role in study design, data collection and interpretation, or the decision to submit the work for publication.

## Author contributions

Ross T Lindsay, Conceptualization, Data curation, Formal analysis, Funding acquisition, Validation, Investigation, Visualization, Methodology, Writing - original draft, Project administration, Writing - review and editing; Sophie Dieckmann, Dominika Krzyzanska, Dominic Manetta-Jones, Investigation; James A West, Cecilia Castro, Formal analysis, Investigation, Methodology; Julian L Griffin, Conceptualization, Resources, Supervision, Funding acquisition, Validation, Visualization, Methodology, Writing - original draft, Writing - review and editing; Andrew J Murray, Conceptualization, Resources, Supervision, Funding acquisition, Validation, Visualization, Methodology, Writing - original draft, Project administration, Writing - review and editing

## Author ORCIDs

Ross T Lindsay (iD) https://orcid.org/0000-0001-7760-613X

## Ethics

Animal experimentation: All experiments conformed to UK Home Office guidelines under the Animals in Scientific Procedures Act, and were approved by the University of Cambridge Animal Welfare and Ethical Review Committee.

## Decision letter and Author response

Decision letter https://doi.org/10.7554/eLife.71270.sa1
Author response https://doi.org/10.7554/eLife.71270.sa2

# Additional files

## Supplementary files

- Supplementary file 1. Pre-ischaemic cardiac function for *Figure 2* hearts.
- Supplementary file 2. Pre-ischaemic contractile function for *Figure 4—figure supplement 2* functional recovery.
- Supplementary file 3. Pre-ischaemic contractile function for *Figure 6* functional recovery.
- Transparent reporting form

## Data availability

The datasets generated during the current study are freely available in the University of Cambridge repository.

The following dataset was generated:

| Author(s) | Year | Dataset title | Dataset URL | Database and Identifier |
| --- | --- | --- | --- | --- |
| Lindsay R, Dieckmann S, Krzyzanska D, Manetta-Jones D, West J, Castro C, Griffin J | 2021 | $\beta$-Hydroxybutyrate Accumulates in the Rat Heart during Low-Flow Ischaemia with Implications for Functional Recovery | https://www.repository.cam.ac.uk/handle/1810/327420 | Apollo - University of Cambridge Repository, 10.17863/CAM.72871 |

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
