## [Decision Letter]

**Acceptance summary:**

This manuscript will be of interest to scientists within the field of cardiac metabolism and acute and chronic heart failure. Although the data support the conclusions of the manuscript, it still unclear whether it may have a translational impact. Nevertheless, it contributes in giving new understanding of the changes in cardiac metabolism under pathological conditions.

**Decision letter after peer review:**

Thank you for submitting your article "β-Hydroxybutyrate Accumulates in the Rat Heart during Low-Flow Ischaemia with Implications for Functional Recovery" for consideration by *eLife*. Your article has been reviewed by 3 peer reviewers, and the evaluation has been overseen by a Reviewing Editor and Matthias Barton as the Senior Editor. The following individual involved in review of your submission has agreed to reveal their identity: Heinrich Taegtmeyer (Reviewer #1).

The reviewers have discussed their reviews with one another, and this letter is to help you prepare a revised submission.

Essential revisions:

1. Previous perfusion studies (Comte 1997, Opie and Owen 1975, as referred to by the author) have reported a net ketone body production in isolated perfused heats. In these experiments, both glucose and fatty acids were present as substrates. Likewise, the initial studies in the present study showed β-OHB accumulation and efflux during ischemia in heart perfused in the presence of both glucose and lipids (triglycerides). In the follow-up studies (using enzyme inhibitors) however, the heart perfusions were done with only glucose. It has been described that both during low-flow ischemia and reperfusion, the reliance on fatty acid oxidation rate are elevated in the heart. This is due to elevated circulating fatty acids and/or altered (less) inhibition of fatty acid oxidation. A brief discussion should be provided on why metabolite levels in inhibitor studies were not assessed pre-ischemia and post reperfusion. Furthermore, the graphs should display absolute metabolite concentrations. This could also be done as a separate supplementary table, depending on the editors' preference. The cartoon figures would be easier to understand if the ketogenic and ketolytic pathways were highlighted in different colours/dashed lines, and if thiolase was represented as a bidirectional flux point.

2. Although the metabolic evaluation is thorough, it should be noted that these do not necessary relate to the flux though the different pathways. Thus, it is unclear why efflux of β-OHB from the heart was not measured when inhibitors were used. Would this not have helped the interpretation of the effect of the inhibitors?

3. Figure 4 and 5 are helpful. However, is is unclear why metabolites that show similar changes (such as acetoacetate and succinyl-CoA) are not indicated by the same arrow and font size.

4. The authors highlight that "urge caution" should be taken when interpreting results using the LDH inhibitor (oxamate) due the use of a relatively high dose. Could you not have used a lower dose and/or an alternative intervention?

5. In Figure 6 it is stated an n=5 heart in each group, while the supplementary table 3 give the pre-ischemic function of n=6.

---

## [Author Response]

Essential revisions:1. Previous perfusion studies (Comte 1997, Opie and Owen 1975, as referred to by the author) have reported a net ketone body production in isolated perfused heats. In these experiments, both glucose and fatty acids were present as substrates. Likewise, the initial studies in the present study showed β-OHB accumulation and efflux during ischemia in heart perfused in the presence of both glucose and lipids (triglycerides). In the follow-up studies (using enzyme inhibitors) however, the heart perfusions were done with only glucose. It has been described that both during low-flow ischemia and reperfusion, the reliance on fatty acid oxidation rate are elevated in the heart. This is due to elevated circulating fatty acids and/or altered (less) inhibition of fatty acid oxidation. A brief discussion should be provided on why metabolite levels in inhibitor studies were not assessed pre-ischemia and post reperfusion. Furthermore, the graphs should display absolute metabolite concentrations. This could also be done as a separate supplementary table, depending on the editors' preference. The cartoon figures would be easier to understand if the ketogenic and ketolytic pathways were highlighted in different colours/dashed lines, and if thiolase was represented as a bidirectional flux point.

We thank the Reviewers for raising these important points.

The decision to perform inhibitor studies with glucose alone was based on experimental data now included in the revised submission (Figure 2—figure supplement 3). Here we found that inclusion of Intralipid in the perfusion buffer did not alter cardiac β-OHB accumulation, compared with hearts perfused with glucose alone. Moreover, in the experiment using oxamate to inhibit LDH, we used glucose as the sole substrate in order to maximise glycolytic flux and enhance acetyl-CoA availability. We have clarified this decision in the revised Methods section (page 16, paragraph 3).

Pre and post ischaemic metabolite levels were not assessed in the inhibitor studies as we did not find β-OHB to be present in either pre-ischaemic or post-ischemic hearts, and as such it was not possible to normalise to this. We have clarified this decision in the revised Methods section (page 17, paragraph 3).

We agree with the Reviewers, that it would have been desirable to see absolute metabolite concentrations, however since mass spectrometry is semi-quantitative, and the relationship between peak size and concentration differs between metabolites and between sample plates, absolute quantification requires addition of standards for each metabolite. We did include a standard ladder for β-OHB in the initial experiments as detailed in the methods section, however to do this for each metabolite, each LC method and each set of experiments would have been prohibitively expensive. Moreover, for some metabolites, a lack of stability may have affected comparisons between the standard ladder and sample concentrations, as the standards would not have been processed by the same method as metabolites extracted from the LV tissue.

We thank the reviewers for their comments on the figures and have implemented the changes suggested.

2. Although the metabolic evaluation is thorough, it should be noted that these do not necessary relate to the flux though the different pathways. Thus, it is unclear why efflux of β-OHB from the heart was not measured when inhibitors were used. Would this not have helped the interpretation of the effect of the inhibitors?

We thank the reviewers for raising this point. We have now included this data in the revised manuscript (Figure 4—figure supplement 1, and Figure 6—figure supplement 1).

3. Figure 4 and 5 are helpful. However, is is unclear why metabolites that show similar changes (such as acetoacetate and succinyl-CoA) are not indicated by the same arrow and font size.

We have now indicated in the figure legends that the boldness of font and arrows correspond to metabolites which accumulated more during ischaemia with than without treatment, and the pathways by which we suggest this may have occurred.

4. The authors highlight that "urge caution" should be taken when interpreting results using the LDH inhibitor (oxamate) due the use of a relatively high dose. Could you not have used a lower dose and/or an alternative intervention?

We thank the Reviewers for raising this. We felt it was important to express the limitations of this particular experiment in our Discussion, particularly regarding the functional data since impaired lactate production would likely also have influenced recovery. We are grateful that the Reviewers recognised our openness in this regard. We agree that it would have been desirable to use an alternative intervention or lower dose, but with limited alternatives available we selected this approach on the basis of previously published work. Despite the limitations, however, we do feel that the results of this experiment were informative, in that the dose of oxamate used did enhance acetyl-CoA availability during ischaemia as we intended, and this resulted in enhanced downstream accumulation of β-OHB supporting this as a candidate pathway.

5. In Figure 6 it is stated an n=5 heart in each group, while the supplementary table 3 give the pre-ischemic function of n=6.

We thank the reviewers for spotting this error and have corrected the Supplementary Tables accordingly.